# Frost Resistance and Microscopic Properties of Recycled Coarse Aggregate Concrete Containing Chemical Admixtures

**DOI:** 10.3390/ma17194687

**Published:** 2024-09-24

**Authors:** Yongyuan Song, Wenjuan Zhou, Chen Zhang, Can Yang

**Affiliations:** 1School of Civil and Transportation Engineering, Beijing University of Civil Engineering and Architecture, Beijing 100044, China; 2108590022111@stu.bucea.edu.cn (Y.S.); zc13306466801@163.com (C.Z.); canyang1105@163.com (C.Y.); 2Engineering Technology Innovation Centre for Construction Waste Recycling, Ministry of Housing and Urban-Rural Development, Beijing 100044, China

**Keywords:** recycled coarse aggregate concrete, mechanical properties, frost resistance, chemical admixture

## Abstract

In order to increase the suitability of coarse recycled concrete aggregates and improve the frost resistance of recycled coarse aggregate concrete, this study aims to investigate the effects of an antifreeze-type water-reducing admixture, air-entraining admixture, and antifreeze admixture on the frost resistance of recycled coarse aggregate concrete. The effectiveness of these admixtures is gauged by the mass loss rate and the relative dynamic modulus of elasticity (RDM). Mercury-impressed porosimetry (MIP), super depth of field microscopy, and scanning electron microscopy (SEM) were employed to characterize the hydration products, microstructure, and pore structure of recycled coarse aggregate concrete, with a view to establishing a connection between the microstructural characteristics and the macro properties and analyzing the micro-mechanism of the improvement effect of frost resistance. The test results demonstrate that the admixtures have a significant impact on the frost resistance of recycled coarse aggregate concrete. In particular, the recycled coarse aggregate concrete with an antifreeze admixture (dosage of 1%) and a water–cement ratio of 0.41 exhibited a mass loss of only 1.23% after 200 freezing and thawing cycles, a relative dynamic modulus of elasticity of up to 93.97%; however, the control group had reached the stopping condition at 150 freeze–thaw cycles with more than 10% mass loss. The recycled coarse aggregate concrete with added antifreeze admixture had a tight connection between the aggregate and the paste and a more pronounced improvement in the pore structure, indicating excellent resistance to frost damage.

## 1. Introduction

With the rapid development of society and large-scale urban renewal, the demand for concrete continues to increase [1]. However, faced with the problem of disposing of large quantities of demolished construction waste, the environment may be exposed to certain risks [2,3]. More than 3 billion tons of construction waste are generated annually by more than 40 countries. China is the world’s largest producer of construction waste with over 1 billion tons [4]. Therefore, the concept of recycled aggregate concrete aims to recycle resources by partially replacing ordinary concrete, meeting social needs, and reducing environmental risks. As a result, studying recycled aggregate concrete has become an important topic in contemporary society [4,5]. However, coarse recycled concrete aggregates are characterized by many initial internal defects, easily cracked, high porosity, high water absorption, etc., because of the large amount of old mortar in recycled aggregates in comparison to natural aggregates [6,7,8,9]. This leads to poor mechanical and durability performance. As a result, recycled aggregate concrete compared to ordinary concrete has certain defects in terms of mechanical properties [10,11,12,13,14,15,16,17]. The growing demand for sustainable construction materials has stimulated considerable research into the incorporation of recycled concrete aggregate (RCA) in concrete composites. The effect of using recycled coarse aggregate (RCA) on the behavior of concrete was experimentally investigated by Elansary et al. [18]. The results show that the use of 30% and 50% coarse aggregate replacement ratios (CARRs) and specific water–cement ratios resulted in better mechanical properties than when using normal coarse aggregate (NCA), while the use of 100% CARR resulted in worse mechanical properties than with NCA.

The damage to concrete structures under the action of a freeze–thaw cycle is a kind of complex fatigue damage; under the action of alternating positive and negative temperatures, the spalling of cement particles on the surface of concrete and the damage of the interface structure of the internal aggregate and the cement tightly bonded are accumulating, and the joint action leads to the freeze–thaw damage of concrete [19]. In cold and frigid regions, the durability of concrete structures is severely compromised and damaged after prolonged freeze–thaw cycles, ultimately leading to building safety problems. Freeze–thaw resistance has long been recognized as an important aspect of concrete durability. Therefore, the study of recycled concrete after a freeze–thaw cycle is particularly important. Freeze–thaw resistance experiments were carried out by Zaharieva et al. [20] on recycled concrete specimens that were subjected to varying degrees of recycled coarse and fine aggregate substitution, as well as distinct freeze–thaw conditions. The findings showed that recycled concrete’s resistance to freeze–thaw is not as good as virgin concrete. Furthermore, the study discovered that a major factor influencing recycled concrete’s ability to withstand freeze–thaw cycles is the water saturation of the surrounding environment. Freeze–thaw cycles have a great influence on the mechanical properties of concrete. The experimental results of Wu and Li [21] also proved that the compressive and tensile strengths of recycled concrete were greatly reduced by the freeze–thaw cycle compared to the original concrete specimens. Furthermore, because of the intricate structure of the system, characterizing the freeze–thaw damage model of recycled coarse aggregate concrete is challenging [22,23]. It is important to note that much of the research is based on techniques and experience used for normal concrete. Studying the whole system of recycled coarse aggregate concrete is more difficult because there are three distinct interfacial transition zones (natural coarse aggregate–new mortar interface, natural coarse aggregate–old mortar interface, and new mortar–old mortar interface) in concrete for coarse recycled concrete aggregates compared to conventional aggregates.

Several studies have demonstrated that adding mineral admixtures, rubber particles, fibers, and admixture can enhance the freeze–thaw resistance of recycled concrete [24,25,26]. According to Wang et al. [27], modified recycled concrete’s ability to withstand freeze–thaw cycles can be enhanced by adding nanosilica and surface-modified basalt fibers. A freeze–thaw damage model was created especially for this type of concrete. This model allows for a detailed analysis of the development process of freeze–thaw damage in materials. The impact of silica fume on the mechanical characteristics and freeze–thaw resistance of concrete made entirely of recycled aggregate was examined by Wang et al. [28]. According to the findings, adding silica fume strengthened the binding between recycled aggregate and cement paste, which enhanced the concrete’s overall performance. The ideal amount of silica fume to add was roughly 13%. In the study by Gokce et al. [29], coarse recycled concrete aggregates produced from non-air-entrained concrete caused poor freezing and thawing resistance in concrete even when the new system had proper air entrainment. Even at lower doses, the freeze–thaw resilience of recycled concrete containing recycled aggregates that do not entrain air was noticeably reduced. The freeze–thaw resistance of recycled concrete may be improved, but not much, by lowering the remaining paste content on the surface of recycled aggregates. In their investigation, Çullu and Arslan [30] investigated how antifreeze affects the physical properties of concrete. After studying a variety of antifreezes, they found that the most effective antifreeze mixture was composed of 30% calcium nitrate and 5% hydroxyethylamine (HEA). However, all forms of antifreeze can negatively affect the physical properties of concrete. The frost resistance of recycled concrete can be improved by incorporating rubber particles, fibers, and admixtures, but, due to the differences in the quality of recycled coarse aggregates, the amount of recycled aggregates, and the test methods, the comparability of the test results is limited, and a lot of research is still needed. The use of admixtures such as water-reducing admixture, air-entraining admixture, etc., can directly improve the ease of recycled concrete, improve fluidity, and is conducive to mechanized construction, thus improving the quality of the project and reducing labor intensity. These admixtures have a clear mechanism of action [31] and can significantly improve the working properties of recycled concrete without changing its basic composition by adjusting the ratio and strength of the concrete. In contrast, the addition of fibers and rubber particles can improve some mechanical properties and durability of concrete, but the amount and type of their additions to the concrete performance of the impact is more complex, and needs more experiments to determine the optimal ratio.

This research investigates the function of antifreeze-type water-reducing, air-entraining, and antifreeze admixture in enhancing recycled coarse aggregate concrete’s freeze resistance. This work examines the fundamental mechanisms underlying microstructural freeze–thaw degradation, with particular emphasis on the impact of admixtures on recycled coarse aggregate concrete’s ability to withstand frost. Combining the classification of crack formation in recycled coarse aggregate concrete with the components of the system generating freeze–thaw damage allows for the diagnosis of the affected areas. This study explores the frost resistance of recycled coarse aggregate concrete in low-temperature environments and proposes improvements to enhance its frost resistance for wider use in engineering construction.

## 2. Materials and Experimental Methods

### 2.1. Materials

The recycled coarse aggregate used in this test is concrete recycled coarse aggregate with a particle size of 5–25 mm, from Beijing Urban Green Source Environmental Protection Technology Co. (Beijing, China). The recycled coarse aggregate has a loose bulk density of 1202 kg/m^3^, a water absorption rate of 4.3%, a density of 2591 kg/m^3,^ and a crushing index of 16%, as per the Chinese code GB/T 25177-2010 [32]. The appearance of the regenerated coarse aggregate is seen in Figure 1.

The fine aggregate used in this test is natural river sand with grain size less than 4.75 mm produced by Beijing Yugou Group Co. (Beijing, China). The physical characteristics of fine aggregate were examined in accordance with Chinese code GB/T 14684-2022 [33]. Its loose bulk density is 1765 kg/m^3^, its density is 2618 kg/m^3^, its water content is 0.3%, and its water absorption is 0.7%. Table 1 depicts the mass of the sieve residue. The natural river sand’s fineness modulus is 2.41, indicating a medium sand with a range of from 2.3 to 3.0, based on the screening findings shown in Table 1.

P·O 42.5 regular Portland cement produced by BBMG (Beijing, China) was used as a binder in this experiment. Fly ash is produced by Hebei Jintaicheng Building Materials Company (Shahe, China) I grade fly ash. The performance index and chemical makeup of the Portland cement (PC) and fly ash (FA) are displayed in Table 2. Three different admixtures were also used: Antifreeze-type water-reducing admixture is provided by High Strength Concrete Company, semi-transparent liquid with 30% solid content and 30% water reduction rate. Anti-freezing type water reducing agent is HZ-2 type anti-freezing type water reducing agent provided by Hebei Hezhong Building Materials Co. (Langfang, China). Air-entraining admixture provided by Urban Green Source Environmental Protection Technology Co., Ltd. (Beijing, China) is alkyl benzene sulfonate air-entraining admixture, semi-transparent liquid. Anti-freezing admixture provided by Hebei Hezhong Building Materials Co., Ltd., HZ-D type anti-freezing admixture, belongs to the compound inorganic salt anti-freezing agent, there are early strength, water reduction components and other composite anti-freezing components. The main chemical composition and main performance indexes of antifreeze are shown in Table 3 and Table 4.

### 2.2. Mix Proportion

The amounts of various materials per cubic meter for the planned concrete with water–cement ratios of 0.56 and 0.41 are displayed in Table 5. Throughout the test, the amount of water-reducing admixture (polycarboxylic acid water-reducing admixture) was changed to control the slump within a 220–240 mm range. As a result of earlier experimental research demonstrating the superior frost resistance of recycled coarse aggregate concrete with a low water–cement ratio, the following was performed in this study:

By controlling the slump, the antifreeze-type water-reducing admixture can adjust the mixing amount. In recycled coarse aggregate concrete with a water cement ratio of 0.56 and 0.41, the antifreeze water-reducer mixing amount was 2.83% and 3.02% (ARRAC-1, ARRAC-2), respectively. ARRAC-1: the antifreeze-type water-reducing admixture uses 9.79 kg per cubic meter of concrete. ARRAC-2: the antifreeze-type water-reducing admixture uses 13.98 kg per cubic meter of concrete.

By varying the dose of the air-entraining admixture, different air content levels can be obtained in the concrete. When the water-to-cement ratio was 0.41, the air content of the recycled coarse aggregate concrete was 4%, 5%, and 6% (AE4%RAC-2, AE5%RAC-2, AE6%RAC-2), respectively. When the ratio was 0.56, the air content was 5% (AE5%RAC-1). The specific quantities of air-entraining admixture used in AE4%RAC-2, AE5%RAC-2, AE6%RAC-2, and AE5%RAC-1 are 0.7 kg, 1 kg, 1.4 kg, and 1.3 kg per cubic meter of concrete, respectively.

At 0.41, antifreeze admixture was added to recycled coarse aggregate concrete in concentrations of 0.5%, 1%, and 1.5% (AF0.5%RAC-2, AF1%RAC-2, AF1.5%RAC-2), respectively. The specific quantities of air-entraining admixture used in AF0.5%RAC-2, AF1%RAC-2, and AF1.5%RAC-2 were 2.2 kg, 4.37 kg, and 6.6 kg per cubic meter of concrete, respectively.

### 2.3. Configuration Process and Maintenance Conditions

Recycled coarse aggregate concrete was prepared using a small mixer, and the recycled coarse aggregate was mixed directly with water without pre-saturated surface dry treatment. The consistency of all recycled coarse aggregate concrete was determined according to the Chinese code GB/T 50080-2016 [36]. After being molded and smoothed, the specimens were wrapped in plastic wrap and kept for from one to two days at 20 ± 5 °C with a relative humidity of at least 50%. After that, they were numbered and demolded, and kept in a conservation room at a temperature of 20 ± 2 °C and relative humidity of 95% until they achieved the required age. The same surroundings were used for the molding and curing of every specimen. Each group of freeze test is three specimens of 100 mm × 100 mm × 400 mm. In the standard curing room, frost resistance specimens should be taken out before the curing age of 24 d and immersed in water at 20 ± 2 °C for 4 d; during the immersion process, the water level is raised to 20–30 mm away from the specimen and the specimen is taken out after 28 d for a frost resistance test.

### 2.4. Testing Methods

GB/T 50081-2019 [37] is the Chinese code for this. Testing the compressive strength is carried out on a specimen measuring 100 mm by 100 mm by 100 mm. After performing maintenance to the designated age, place the apparatus onto the hydraulic universal testing machine. Maintain a consistent and uninterrupted loading procedure at a pace of from 0.3 to 1.0 MPa/s.

This study was conducted in accordance with the Chinese code GB/T 50082-2009 [38], with the objective of assessing the frost resistance of recycled coarse aggregate concrete. This was achieved through the utilization of the mass loss rate and relative dynamic modulus of elasticity test. Relative dynamic modulus of elasticity test using the DT-10 dynamic elasticity meter produced by Tianjin Zhongshi Science and Technology Instrument Company (Tianjin, China). The specimen’s initial mass and dynamic modulus of elasticity are measured when it is 28 d old. Every freeze–thaw cycle ought to be finished in from two to four hours, and the amount of time spent thawing should be greater than one-fourth of the overall duration. The mass and dynamic modulus of elasticity are measured on specimens placed in the freeze–thaw chamber at regular intervals until the test is concluded. Upon completion of 200 cycles, the test is terminated. Alternatively, the relative dynamic modulus of elasticity may be less than or equal to 60%, or the mass loss may be greater than or equal to 5%.

The mass loss rate and relative dynamic modulus of elasticity (RDM) have been calculated using Equations (1) and (2).
(1)$$W_{n}=\frac{M_{0}-M_{n}}{M_{0}}\times 100\%$$
(2)$$P_{n}=\frac{f_{n}^{2}}{f_{0}^{2}}\times 100\%$$
where $W_{n}$ and $P_{n}$ represent the mass loss rate and RDM, respectively, while $M_{0}$ and $f_{0}$ denote the mass and transverse fundamental frequency before undergoing freeze–thaw cycles. Similarly, $M_{n}$ and $f_{n}$ represent the mass and transverse fundamental frequency after n freeze–thaw cycles.

The samples for scanning electron microscopy were blocks of ordinary cured broken concrete with a diameter of ≤1 cm and a thickness of <1 cm. The samples were gold-sprayed, and images were taken at different magnifications using a ZEISS GeminiSEM 300 scanning microscope from Oberkochen, Germany. The specimens for the mercury compression experiments had dimensions of 20 mm in diameter and 20 mm in height. The MIP samples were placed in anhydrous ethanol for at least 24 h to dehydrate and terminate hydration. Subsequent to the drying of the samples in an oven at a temperature of 45 °C for a period of 24 h, until a constant mass was reached, the samples were allowed to undergo a process of evaporation, whereby any free water and alcohol present were removed. The internal pore structure of the blocks was tested using a high-performance, fully automated mercury piezometer (AutoPore Iv 9510, from McMurray Tic Instruments, Norcross, GA, USA) in pore structure measurement mode.

## 3. Results and Discussion

### 3.1. Compressive Strength

The effect of several admixtures on recycled concrete’s compressive strength is shown in Table 6. The present study sought to investigate the influence of diverse admixtures on the compressive strength of concrete. The data indicate that concrete with a water–cement ratio of 0.56 (RAC-1) exhibits a lower compressive strength than concrete with a ratio of 0.41 (RAC-2), irrespective of age. The early strength of concrete is less affected by the inclusion of antifreeze water-reducing admixture than by RAC-1 and RAC-2 with polycarboxylic acid water-reducing admixture. However, it significantly reduces the strength of concrete in the later stages, and the strength seems to be significantly reduced at later ages, with the reduction ranging from 5% to 8%; the air-entraining admixture concretes (AE5%RAC-1, AE4%RAC-2, AE5%RAC-2, AE6%RAC-2) had relatively low compressive strength at all ages. Concrete’s compressive strength dropped with increasing air content at all ages, suggesting that air-entraining admixture could have an impact on compressive strength. When concrete antifreeze admixtures (AF0.5%RAC-2 and AF1%RAC-2) were added, the compressive strength of the resulting mixture remained quite high over all age groups. This implies that adding specific concrete antifreeze admixtures could boost the compressive strength. In conclusion, when antifreeze and antifreeze-type water-lowering admixture are employed, the compressive strength of concrete is very high at early ages; nevertheless, when air-entraining admixtures are utilized, the compressive strength of concrete is relatively low at all ages.

### 3.2. Freeze Resistance Performance

#### 3.2.1. Antifreeze-Type Water-Reducing Admixture

The impact of antifreeze-type water-reducing admixture on the mass loss rate and relative dynamic modulus of elasticity of recycled coarse aggregate concrete is illustrated in Figure 2. Figure 2a depicts the mass loss rate of recycled coarse aggregate concrete (RAC) incorporating an antifreeze-type water-reducing admixture. Following 100 freeze–thaw cycles, the mass loss rate in the RAC-1 group was 11%, whereas, in the RAC-2 group, it was just 0.76%. The water-to-cement ratio exerts a pronounced influence on the mass loss rate of recycled coarse aggregate concrete. The mass loss rates of ARRAC-1 and ARRAC-2 with an antifreeze-type water-reducing admixture were 4.73% and 4.46% at 200 freeze–thaw cycles, respectively. These rates remained relatively stable as the number of freeze–thaw cycles increased. The use of an antifreeze-type water-reducing admixture in recycled coarse aggregate concrete did not significantly influence the mass loss rate, regardless of the water-to-cement ratio.

The addition of antifreeze-type water-reducing admixture substantially lowered the decline in the relative dynamic modulus of elasticity of the recycled coarse aggregate concrete, as illustrated in Figure 2b. At 100 and 150 freeze–thaw cycles, respectively, RAC-1 and RAC-2 reached the stop–cycle condition, and their relative dynamic modulus of elasticity dropped to 70% and 70.1%, respectively. However, the freeze–thaw cycle may still be maintained by ARRAC-1 and ARRAC-2, whose relative dynamic modulus of elasticity is 81.58% and 95.52% at 200 times, respectively. In recycled concrete freeze–thaw cycles, antifreeze-type water-reducing admixtures are more effective in minimizing the loss of relative dynamic modulus of elasticity.

After adding an antifreeze-type water-reducing admixture, the recycled coarse aggregate concrete’s resistance to frost was greatly increased. Both spatial resistance and electrostatic repulsion or adsorption can be double superposed in the cement dispersion process because of the polycarboxylic acid water-reducing admixture. However, spatial resistance predominates [39]. The water-reducing components contained in the water-reducing agent will form an adsorption film on the surface of the cement particles. This effect reduces the amount of supercooled water in hardened concrete, preventing icing and reducing the number of capillary pores. The pore structure of the concrete is refined and its densification is improved [40,41,42]. On the other hand, the air-entraining component contained in this water reducer introduces many tiny air bubbles into the concrete and increases the frost resistance of the concrete [42].

#### 3.2.2. Air-Entraining Admixture

Figure 3 illustrates the impact of an air-entraining admixture on the relative dynamic modulus of elasticity and mass loss rate of recycled coarse aggregate concrete. This study reveals that, in comparison to non-air-entrained recycled coarse aggregate concrete, air-entrained recycled coarse aggregate concrete exhibited a significantly reduced mass loss rate (Figure 3a). Moreover, even after 100 freeze–thaw cycles, the mass loss rate of air-entrained recycled coarse aggregate concrete remained low. The mass loss rate of recycled coarse aggregate concrete with air entrainment increased significantly after 100 cycles of freeze–thaw, although it remained below that of recycled coarse aggregate concrete without air entrainment. The mass loss rate of recycled coarse aggregate concrete is significantly influenced by the water-to-cement ratio. After 150 cycles, AE5%RAC-1 meets the freeze–thaw test stop condition, and, after 200 cycles, AE5%RAC-2 has the lowest mass loss rate—just 3.76%—a considerable improvement on the benchmark group.

Figure 3b illustrates that the relative dynamic modulus of elasticity of recycled coarse aggregate concrete with an air-entraining admixture declines more slowly as the number of freeze–thaw cycles increases in comparison to concrete without an air-entraining admixture. Moreover, the water-to-cement ratio influenced the relative kinetic elastic modulus of recycled coarse aggregate concrete. Concrete with a ratio of 0.41 exhibited a greater relative kinetic elastic modulus than that of 0.56. Following 150 freeze–thaw cycles, the AE5%RAC-2 group exhibited a relative dynamic modulus of elasticity of 95.23%, in comparison to 70.1% for the RAC-2 group. Moreover, the relative dynamic modulus of elasticity loss was found to be smaller in the AE5%RAC-2 group compared to the other groups.

In the mixing process of concrete, due to the same charge of the directional adsorption layer, the formation of a large number of mutually exclusive and uniformly distributed bubbles, and most of the air-entraining admixture are the anionic surfactant, which can play a protective role for the bubbles, so the bubbles are not easy to burst and can be stabilized within a certain period of time [43,44]. Within a certain range, the more air holes are introduced, the better the freeze–thaw resistance of the concrete is, but, beyond a certain range, too high an air content will reduce the compactness of the concrete system, causing a significant loss of compressive strength and the resistance to freeze–thaw damage ability to decline [45]. Therefore, the compressive strength of the recycled concrete is significantly reduced at 6% air content, and the effect of improving frost resistance is weakened.

#### 3.2.3. Antifreeze Admixture

The impact of antifreeze on the relative dynamic modulus of elasticity and mass loss rate of recycled coarse aggregate concrete is depicted in Figure 4. The rate of mass loss in recycled coarse aggregate concrete was considerably decreased by the inclusion of antifreeze. The mass loss rate of recycled coarse aggregate concrete was noted at various antifreeze doses, as shown in Figure 4a. The findings showed that the comparable mass loss rates at antifreeze admixture concentrations of 0.5%, 1%, and 1.5% were 2.35%, 1.85%, and 1.23%, respectively. It is noteworthy that there was not much of a difference in the mass loss rate between the various antifreeze dosages. Furthermore, after 200 freeze–thaw cycles, the study assessed the mass loss rate of recycled concrete. It is interesting to note that the recycled concrete with a 1% antifreeze dosage had the lowest mass loss rate. This implies that recycled concrete’s resilience to freeze–thaw cycles can be increased by applying a 1% antifreeze dosage.

The relative dynamic modulus of elasticity for examining various antifreeze admixture doping doses is displayed in Figure 4b. The findings indicate that the relative dynamic modulus of elasticity for concentrations of 0.5%, 1%, and 1.5% varies very little. Nonetheless, it was discovered that, at the same concentrations, the relative dynamic modulus of elasticity at 200 freeze–thaw cycles was 92.72%, 93.97%, and 91.6%. The results indicate that the relative kinetic elastic modulus was higher when the antifreeze admixture doping concentration was 1%. This suggests that the improvement effect on the elastic modulus was more pronounced in the 1% antifreeze admixture doping group compared to the other concentrations tested. These findings highlight the potential benefits of using a 1% antifreeze doping concentration in enhancing the relative dynamic modulus of elasticity under freeze–thaw conditions.

Concrete’s ability to withstand icing and expansion is enhanced by the addition of antifreeze admixture, which can also speed up the formation of the material’s pore structure, increase early strength, decrease the free water content, and encourage alterations in the ice crystal structure [31,46]. For recycled fine aggregate concrete to be more resistant to freeze–thaw cycles, antifreeze chemicals are essential. The overall effect is beneficial, even though there can be a minor worsening in the mass loss rate during the early phases of the freeze–thaw cycle. When antifreeze admixtures are used, the relative loss of dynamic elastic modulus is significantly reduced. This effect intensifies as the number of freeze–thaw cycles increases. As a result, the benefits of adding antifreeze admixture to recycled coarse aggregate concrete gradually become clearer.

### 3.3. SEM

#### 3.3.1. Interface Structure

Recycled concrete contains a large number of uneven pores and interface cracks, as shown in Figure 5a. Additionally, there are cracks between the aggregate and concrete when it is not under pressure, as shown in Figure 5b. During the repeated freeze–thaw cycles, the pore walls of these tiny pores experience pressure, causing the interfacial structure to become even looser. Consequently, the tiny pores start to connect with each other, leading to the formation of cracks. The detachment of the mortar layer on the surface of the specimen, exposing the underlying aggregate, is a macroscopically evident phenomenon. This diminishes the frost resistance of the concrete.

After the addition of the antifreeze-type water-reducing admixture, the densification of the interfacial transition zone was significantly improved, resulting in a noticeable reduction in cracks (as depicted in Figure 5c,d). However, despite these improvements, the slurry portion still exhibited a relatively rough texture, with the presence of irregular pores. As a consequence, the compressive strength and resistance to chlorine ion penetration performance experienced a slight decrease. After the air-entraining admixture is mixed, it introduces a significant number of tiny bubbles into the mixture. These bubbles play a crucial role in refining the pore structure of the mortar and homogenizing the structure of both the new mortar and the interface zone. Despite these improvements, the recycled coarse aggregate itself possesses inherent structural defects, leading to the presence of large cracks in the interface transition zone. This can be observed in Figure 5e,f. Furthermore, the absence of new generation hydration products between the new mortar and the aggregate indicates that these cracks are most likely formed during the crushing process of the recycled aggregate. After the addition of antifreeze admixture, the newborn cement structure undergoes significant improvements. Despite the presence of holes, the overall density of the structure is enhanced, resulting in fewer microcracks. This can be observed in Figure 5g,h, where the slurry is tightly bonded to the aggregate, indicating a strong and cohesive bond. The addition of antifreeze admixture also accelerates the hydration and hardening process of the concrete, leading to improved early strength.

#### 3.3.2. Hydrated

Figure 6a shows that cracking between the hydration products and aggregates of the RAC-2 group can have a significant impact on the overall strength and durability of the concrete. These cracks can weaken the bond between the aggregates and the hydration products, leading to reduced structural integrity. Furthermore, the absence of fibrous and clustered products in the RAC-2 group suggests that the concrete may lack reinforcement and cohesion. Their absence in this case could contribute to the reduced properties of the concrete. The irregular granular materials with loose connections that were observed in the RAC-2 group can further compromise the overall quality of the concrete. The recycled coarse aggregate concrete with antifreeze-type water-reducing admixture is still visible as incompletely hydrated fly ash particles in Figure 6c, with more clustered and radial products of hydration products water C-S-H gel, and flaky crystals and fewer needle and rod AFts are visible at high magnification in Figure 6d. In addition to the well-developed plate-like Ca(OH)_2_ crystals and worm-like C-S-H gel, Figure 6e,f also reveals some other interesting features of the hydration products of the concrete with the air-entraining admixture. The microscopic images show that the hydration products are dense, indicating a strong bond between the cement particles. This denseness is further enhanced by the presence of a few needles and rod-shaped AFt crystals in the middle. These crystals contribute to making both the interfacial transition zone and the internal structure of the cement stone denser. The use of antifreeze admixture in concrete can enhance the early strength of the material. This means that the concrete gains sufficient strength in its early stages, enabling it to bear loads and resist external forces sooner In Figure 6g,h, a significant number of lamellar crystals can be observed. They strengthen and extend the concrete’s general durability, increasing its resistance to damage and cracking.

### 3.4. Super-Depth-of-Field Microscope

In Figure 7, it is evident that the red area of RAC-2 concrete exhibits a higher density and population of pores. However, when an antifreeze-type water-reducing admixture and antifreeze admixture are added to the mix, the pores become more scattered, and their overall number decreases. On the other hand, when an air-entraining admixture is introduced, the number of pores increases. This is due to the formation of air bubbles, which disrupt the connectivity between the pores. Based on the pore data presented in Table 7, it is evident that the total pore number of RAC-2 was initially 656,932. However, after the addition of antifreeze admixture and antifreeze-type water-reducing admixture, the pore number decreased by 26.4% and 26.8%, respectively. This reduction in the pore number indicates that the frost resistance performance of concrete can be enhanced by employing these three methods.

### 3.5. Mercury Injection

Figure 8 shows the differential curve of the concrete pore size distribution. The most available pore size is the pore size corresponding to the peak, the pore with the highest probability of occurrence, and only holes larger than this pore size can form a connected pore channel [47,48]. The addition of antifreeze-type water-reducing admixture, compound antifreeze, or air-entraining admixture to RAC-2 resulted in a reduction in the most available pore size. Specifically, the most available pore size decreased to 41.28 nm, 40.27 nm, and 40.25 nm, representing a reduction of 6.82%, 9.10%, and 9%, respectively. Comparing the pore size refinement of ARRAC-2, AFRAC-2, and AERAC-2 with RAC-2, it was observed that the first peak shifted towards smaller pore sizes. Additionally, the range of pore sizes between 1 and 10 nm increased. These changes in pore size distribution have the potential to optimize the overall pore size distribution in concrete, leading to a refined pore structure and a reduction in larger pore sizes [47]. Therefore, its connectivity deteriorates, the pore volume fineness increases, and resistance to freeze–thaw cycles increases. This is similar to Jin et al. [49] findings that the freeze–thaw resistance of concrete mainly depends on the hydration products and pore composition, and a dense concrete structure, which is conducive to the improvement in the freeze resistance of concrete.

Table 8 shows the pore structure parameters of recycled coarse aggregate concrete. The total pore volume, total porosity, average pore size, and maximum usable pore size of the concrete decreased with the addition of antifreeze-type water-reducing admixture and antifreeze admixture. It is noteworthy that the maximum available pore diameter and total pore volume actually decreased with the addition of air-entraining admixture despite the increase in total porosity. This suggests that although the air-entraining admixture introduced more pores into the concrete, the size of these pores became smaller.

Mean pore size is an important parameter of the concrete pore structure [50]. The complexity of the spatial distribution of micropores in concrete is closely tied to the average pore size. In general, as the average pore size decreases, the complexity of the distribution increases. This means that smaller average pore sizes result in a higher degree of intricacy in how the micropores are dispersed throughout the concrete matrix. When an anti-freezing type of water-reducing admixture or antifreeze admixture is added to the concrete mix, it has a significant impact on the pore structure. Specifically, it reduces the average pore size and decreases the number of large pores. As a result, the number of micropores increases, leading to an overall densification of the concrete material. This densification is beneficial for enhancing the anti-freezing performance of the concrete. By reducing the average pore size and increasing the number of micropores, the concrete becomes more resistant to the damaging effects of freezing and thawing cycles.

The critical pore size is the pore size corresponding to the location where the cumulative pore volume begins to increase substantially; the maximum pore level that can connect the larger pores, and pores larger than the critical pore size will affect permeability [51]. From the curves in Figure 9, it can be inferred that the addition of antifreeze-type water reducer admixture or antifreeze admixture greatly reduces the critical pore size.

The addition of an antifreeze-type water-reducing admixture and antifreeze to recycled concrete resulted in a decrease in various pore characteristics. There was a decrease in the total porosity, total pore volume, and average, critical, and maximum pore diameters. This can be attributed to the properties of the antifreeze-type water-reducing admixture and antifreeze, which likely contribute to densifying the concrete structure and reducing the size of the pores. On the other hand, when an air-entraining admixture was added, the effects on the pore characteristics were slightly different. Even with the addition of the water-reducing admixture and antifreeze of the antifreeze type, the overall porosity decreased, but not by as much. Furthermore, there was a small rise in the average pore diameter. On the other hand, the maximum achievable pore diameter and the overall pore volume shrank. This can be explained by the fact that the air-entraining admixture introduces stable closed and uniform small bubbles into the concrete, which inhibit the formation of large pores. As a consequence, the concrete’s pore structure becomes more refined and its pore curvature rises. Overall, the frost resistance of recycled coarse aggregate concrete is enhanced by the use of external admixture.

## 4. Conclusions

In order to improve the frost resistance of recycled coarse concrete, the authors chose two water–cement ratios, 0.41 and 0.56, which are representative of current research on recycled concrete aggregates. In addition, the physical properties and frost resistance of the recycled concrete were tested, and the micro-interfaces, hydration products, and pore structure of the recycled concrete were analyzed by SEM, ultra-field microscopy, and piezomercury tests. The results of these tests led to a number of conclusions.

Concrete with antifreeze and antifreeze-type water-reducing admixtures had relatively high compressive strength at early ages, while concrete with air-entraining admixtures had relatively low compressive strength at all ages.When adding antifreeze-type water-reducing admixture, air-entraining admixture, and antifreeze admixture to concrete, the freeze–thaw resistance of recycled coarse aggregate concrete improves. After more than 100 freezing and thawing cycles, the mass loss rate and relative dynamic modulus of elasticity loss of the recycled concrete with air-entraining admixture were significantly reduced compared with the baseline group, in which the recycled concrete with a water–cement ratio of 0.41 and an air-entraining admixture dosage of 5% had the best freezing-resistant effect. In recycled concrete with different types of admixtures, following 100 freeze–thaw cycles, recycled coarse aggregate concrete with a water-to-cement ratio of 0.41 and antifreeze (1% dosage) has the best 28 d compressive strength, the lowest mass loss rate, and a clear reduction in the loss of relative dynamic modulus of elasticity. The aggregate is closely bonded with the paste, and the pore structure has been obviously improved, which shows excellent ability to improve the antifreeze performance.Compared with the control group, the recycled concrete with antifreeze-type water-reducing admixture, air-entraining admixture, and antifreeze admixture has a denser internal structure, the aggregate and paste are tightly bonded, the number of cracks and pores is reduced, and its freezing resistance has been significantly improved.

The crucial role of admixtures in enhancing the frost resistance of concrete using recycled coarse aggregate is emphasized in the study. When antifreeze-type water-reducing admixture, air-entraining admixture, and anti-freeze admixture are added to concrete, the frost resistance of concrete can be significantly improved, which is essential for maintaining the long-term stability and security of concrete structures in cold climates. This will promote the usage of recycled coarse aggregate concrete in colder climates on a larger scale.

## Figures and Tables

**Figure 1 materials-17-04687-f001:**
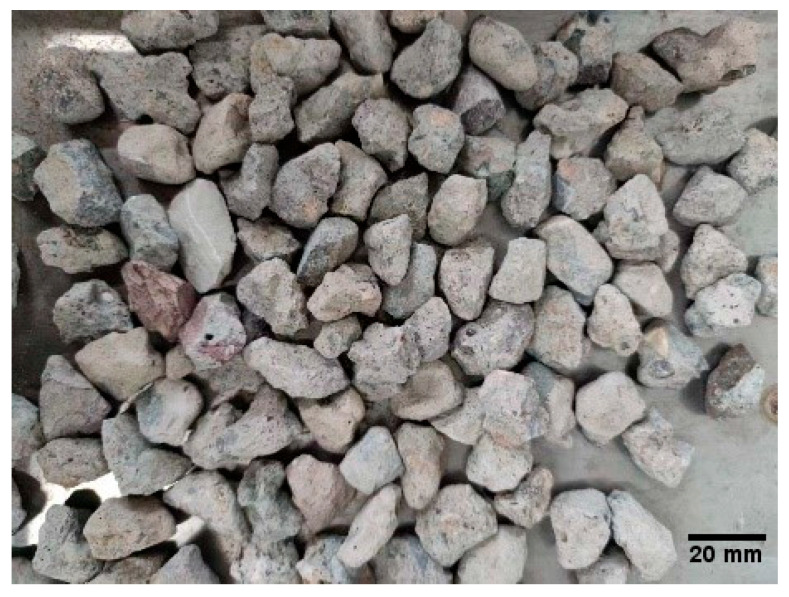
The coarse recycled concrete aggregate.

**Figure 2 materials-17-04687-f002:**
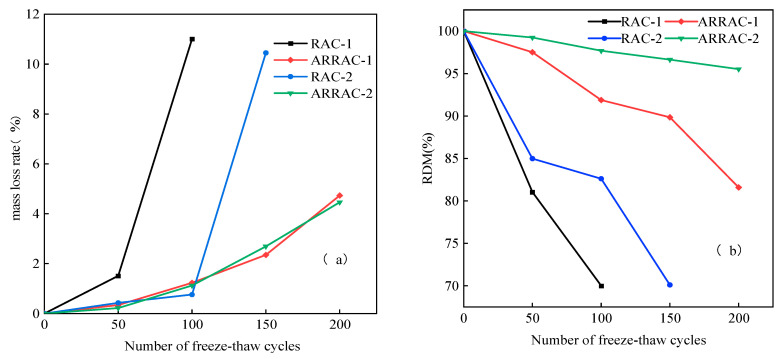
Recycled coarse aggregate concrete with antifreeze-type water-reducing admixture: (**a**) mass loss rate; (**b**) relative dynamic modulus of elasticity.

**Figure 3 materials-17-04687-f003:**
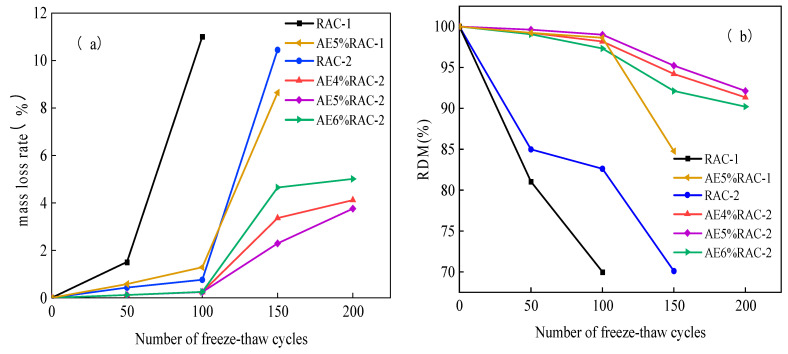
Recycled coarse aggregate concrete with air-entraining admixture: (**a**) Mass loss rate; (**b**) relative dynamic modulus of elasticity.

**Figure 4 materials-17-04687-f004:**
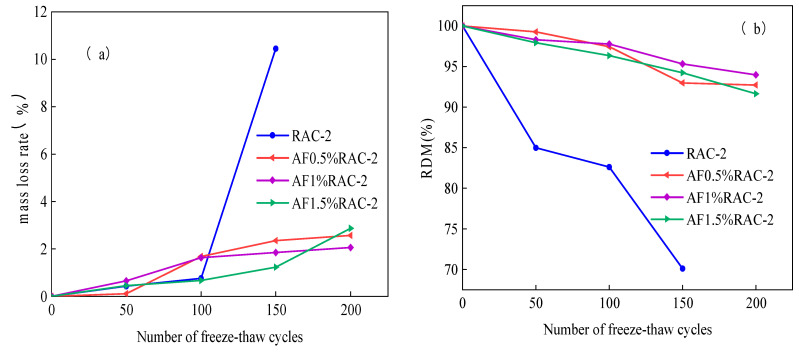
Recycled coarse aggregate concrete with antifreeze admixture: (**a**) Mass loss rate; (**b**) relative dynamic modulus of elasticity.

**Figure 5 materials-17-04687-f005:**
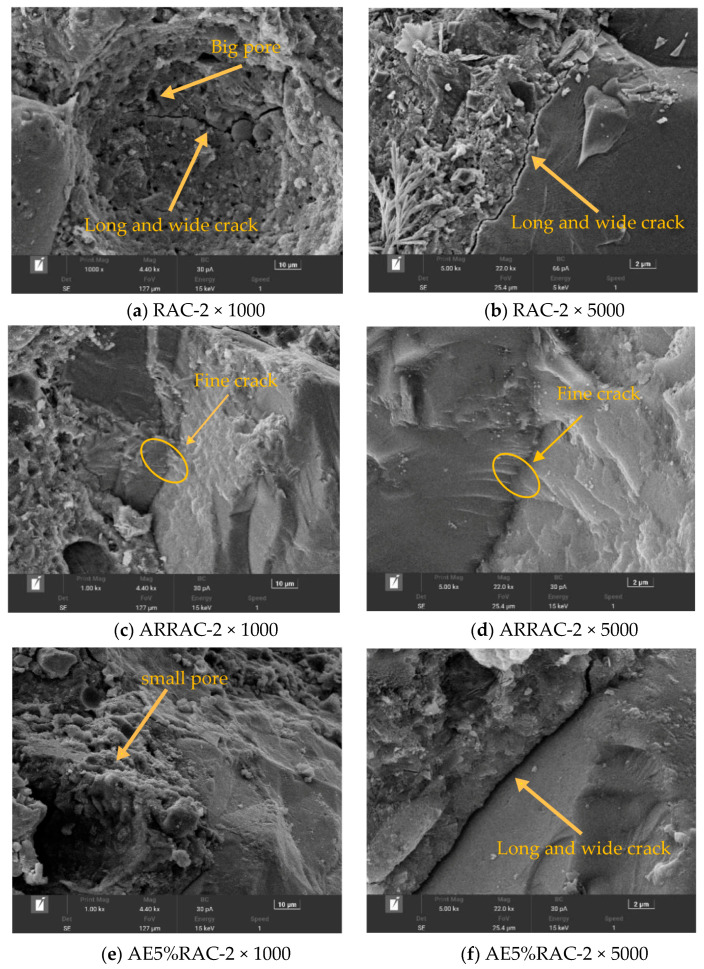
Interface structure SEM image.

**Figure 6 materials-17-04687-f006:**
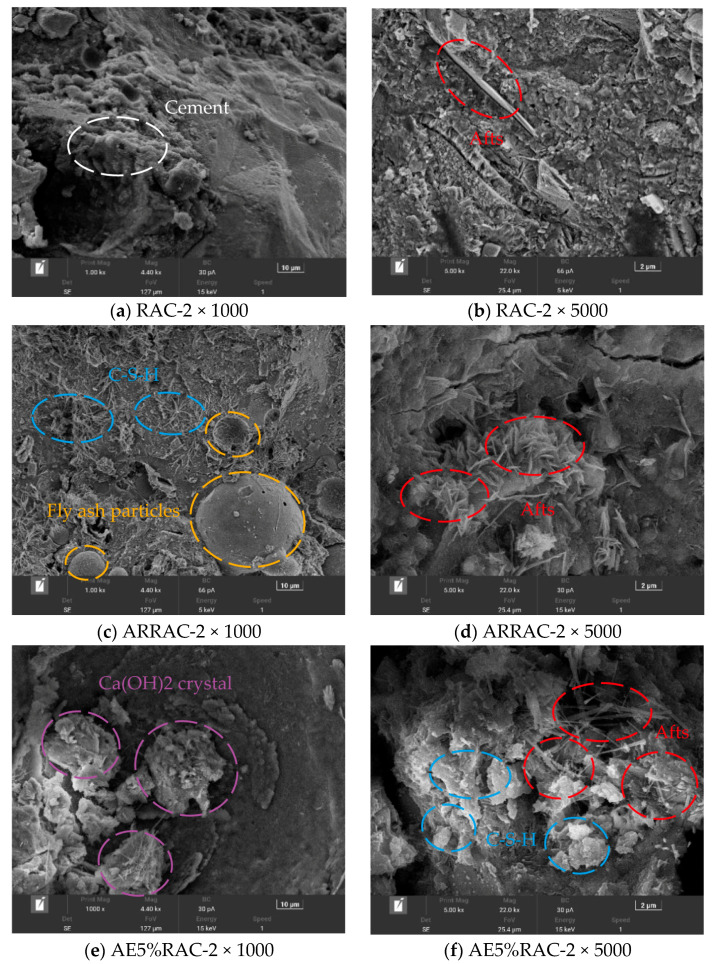
Hydrated SEM image.

**Figure 7 materials-17-04687-f007:**
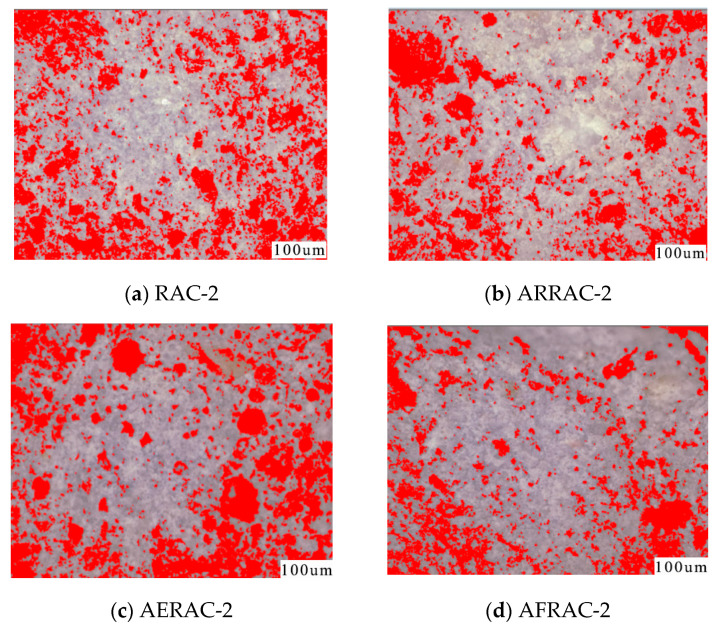
Pore distribution of recycled coarse aggregate concrete.

**Figure 8 materials-17-04687-f008:**
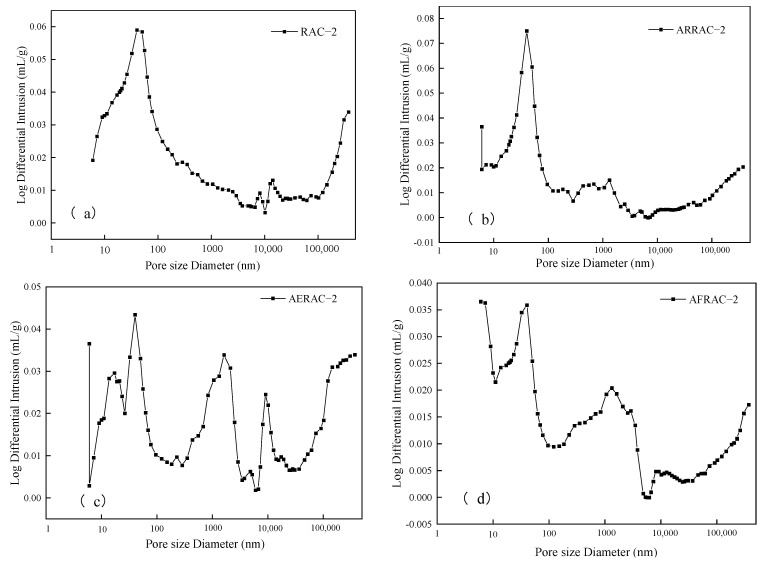
Pore size distribution differential curve. (**a**) RAC-2; (**b**) ARRAC-2; (**c**) AERAC-2; (**d**) AFRAC-2.

**Figure 9 materials-17-04687-f009:**
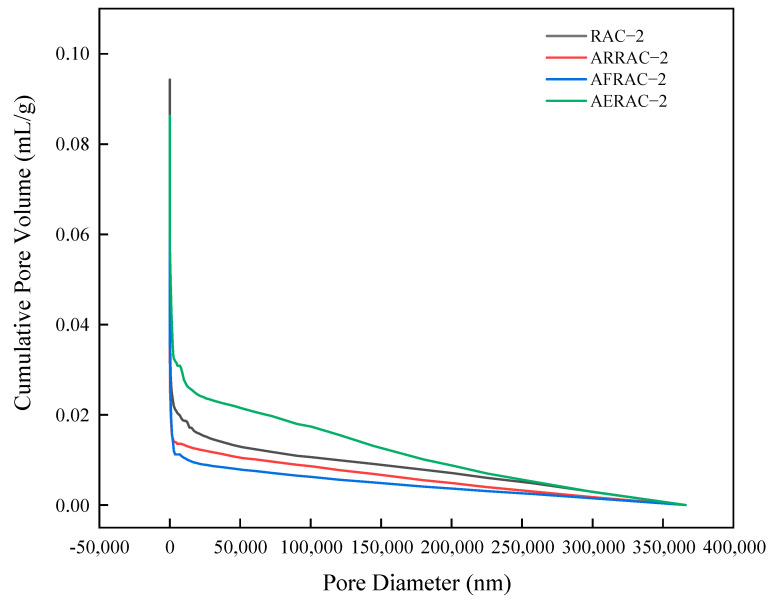
Volume distribution of cumulative holes of recycled coarse aggregate concrete.

**Table 1 materials-17-04687-t001:** Results of fine aggregate sieving tests.

Fine Aggregate	Amount of Residue [g]					
	2.36 mm	1.18 mm	0.6 mm	0.3 mm	0.15 mm	Sieve Bottom
Natural river sand	111	50.7	128	77.4	105.1	27.8

**Table 2 materials-17-04687-t002:** Chemical composition and physical characteristics of PC and FA.

	PC	FA
	chemical analyses, percentage	
SiO_2_ (%)	20.68	37.65
Al_2_O_3_ (%)	7.76	44.06
Fe_2_O_3_ (%)	3.26	9.79
CaO (%)	60.09	6.19
MgO (%)	3.01	0.28
K_2_O (%)	-	0.58
SO_3_ (%)	3.31	0.39
Insoluble residue	1.89	1.06
Pozzolanic activity index ASTM C618 [34]-ASTM C311 [35]		0.81
Compressive strength (MPa) days		
3 d	20.2	
28 d	52.9	
Flexural strength (MPa) days		
3 d	4.7	
28 d	9.1	
	Physical tests	
Density (g/cm^3^)	3.2	2.19
Passing 45 µm, percent	85	90.3
Initial setting time (min)	155	
Final setting time (min)	305	

**Table 3 materials-17-04687-t003:** Main chemical composition of antifreeze (%).

Al_2_O_3_	CaO	Br	Y_2_O_3_	Pd	Ag_2_O	Other
59.3	14.53	13.32	6.38	3.31	0.99	2.17

**Table 4 materials-17-04687-t004:** Key performance indicators for antifreeze.

Solid Content/%	Density g/cm^3^	pH	Water Reduction Rate/%	Air Content/%	28 d Compressive Strength/MPa
37.12	1.193	4.02	30	3.3	148

**Table 5 materials-17-04687-t005:** Mix proportions per cubic meter of the reference concretes.

Water Cement Ratio	Cement (kg/m^3^)	Fly Ash (kg/m^3^)	Sand (kg/m^3^)	RCA * (kg/m^3^)	Water (kg/m^3^)
0.56	242	104	823	1006	194
0.41	306	131	750	1000	181

* RCA: recycled coarse aggregate.

**Table 6 materials-17-04687-t006:** Compressive strength of recycled concrete at different ages.

Admixture	Water Cement Ratio	Specimen No.	Compressive Strength/MPa			
			3 d	7 d	28 d	90 d
	0.56	RAC ^1^-1	18	22	30	36
	0.41	RAC-2	28.5	29	44	50
Antifreeze-type water-reducing admixture	0.56	AR ^2^RAC-1	19	18.5	28.5	33
	0.41	ARRAC-2	26	30	41	47
Air-entraining admixture	0.56	AE ^3^5%RAC-1	12.5	12.5	19.5	26
	0.41	AE4%RAC-2	22	24	36	41.5
	0.41	AE5%RAC-2	20	20	32.5	34.5
	0.41	AE6%RAC-2	11.5	12.5	16	21
Antifreeze admixture	0.41	AF ^4^0.5%RAC-2	29	29.5	44.5	43
	0.41	AF1%RAC-2	31	32	45	46
	0.41	AF1.5%RAC-2	23	25	38	43

^1^ RAC stands for Recycled Coarse Aggregate Concrete; ^2^ AR stands for antifreeze-type water-reducing admixture; ^3^ AE stands for air-entraining admixture; ^4^ AF stands for antifreeze admixture.

**Table 7 materials-17-04687-t007:** Pore data of recycled coarse aggregate concrete.

Number	RAC-2	ARRAC-2	AERAC-2	AFRAC-2
Min area/nm^2^	10	10	10	10
Max area/nm^2^	126,643	84,559	126,778	109,906
Mean	383.3	372.7	630	431.8
Total Pore Number	656,932	480,725	675,622	483,537

**Table 8 materials-17-04687-t008:** Pore structure parameters of recycled coarse aggregate concrete.

No.	RAC-2	ARRAC-2	AFRAC-2	AERAC-2
Total porosity/%	17.27	12.70	13.37	17.05
Total pore volume/mL/g	0.09	0.07	0.07	0.08
most probable pore diameter/nm	44.30	41.28	40.27	40.25
Average pore size/nm	38.35	36.67	33.26	62.61

## Data Availability

The original contributions presented in this study are included in the article, further inquiries can be directed to the corresponding author.

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
