# Peer review of "Frost Resistance and Microscopic Properties of Recycled Coarse Aggregate Concrete Containing Chemical Admixtures"

_materials, 2024, doi:10.3390/ma17194687_

Round 1

Reviewer 1 Report

Comments and Suggestions for Authors

Titles should avoid prefaces such as "A study on..." as they tend to weaken the impression of the paper. Presuming the paper makes significant contributions to the subject area, "Frost Resistance and Microscopic Properties of Recycled Coarse Aggregate Concrete Containing
External Additives" sounds better.

Still on the title, what is an "external additive" as opposed to an "internal additive"?

At the end of the abstract it would be helpful to note the mass loss of the control specimen. Otherwise it is hard to gage the performance of the specimen with anti freezing additive.

Line 36. It is technically imprecise to say "However, recycled coarse aggregates are characterized by ... rapid cracking". What does that mean? The internal porosity and defects weaken the material but can possibly make it less brittle.

line 44 This needs to be written more clearly "the surface particles spalling and internal structural damage continue to accumulate and act together"

lines 45-47 is not true, in general. Many concretes if properly designed do not "ultimately rendering the building unfit for normal use."

line 47 "Freeze–thaw resistance has become an important aspect of the durability of concrete" sounds strange since the freeze-thaw resistance was known to be important long decades ago. It was not discovered by Zhang and colleague in 2022.

On same note "Freeze–thaw cycles have a major impact on the mechanical properties of concrete, according to Wu and Li" is misleading. The bad influence on mechanical properties has been known and widely studied over the past many decade. It was not newly found by Wu and Li in 2017. See the followings. Powers, T.C., “A Working Hypothesis for Further Studies of Frost resistance of Concrete,” ACI Journal, Proceedings, vol. 41, no. 4, February 1945, pp. 245–272.

Powers, T.C., with discussion by Willis, T.F., “The Air Requirement of Frost Resistant Concrete,” Proceedings, Highway Research Board, vol. 29, 1949, pp. 184–211; Bulletin No. 33, Research and Developments Laboratories of the Portland Cement Association.

Line 77 it is not clear why comparison is made with fresh concrete? "exhibited freeze–thaw resilience comparable to that of fresh concrete."

The meaning and point of lines 82-87 needs clarifying.

The following text does not read well and also seems unnecessary. It could be removed. "The majority of the research focuses on the study of recycled concrete’s freeze resistance; yet, it is vital to consider how admixtures can increase recycled concrete’s freeze resistance. This study aims to remedy this shortcoming and enhance knowledge of the function of antifreeze-type water-reducing, air-entraining, and antifreeze additives in enhancing recycled coarse aggregate concrete’s freeze resistance." >> and replaced by simply "This research investigates the function of antifreeze-type water-reducing, air-entraining, and antifreeze additives in enhancing recycled coarse aggregate concrete’s freeze resistance."

For Table 1 the central heading can be "Amount of Residue /%". What do the numbers in the table mean? There is 111%, 50.7%, 128% and on.

Table 2 the meaning of PC and FA are not defined anywhere.

Figure 2 presents common knowledge type information. It does not add to the paper and should be removed.

The captions for Tables 7 and 8 should tell the method used for measurement.

In Table 8 how can the average pore size of AERAC be larger than the maximum pore size? Also the averages are close to the maximums. Is that correct?

For Figures 8, 9 and 10 also the technique for measurement should be included the figure captions.

For conclusion 2 "When additives are added," needs expansion. What additives are being added?

Conclusion 3 says "After the recycled concrete was subjected to frost resistance improvement, the aggregate and paste were tightly bonded, the number of cracks and pores were reduced, and the internal densification of the concrete was increased. " It seems like cause and effect are reversed. The latter parts of the sentence should lead to the front part, not the other way around.

At end of the paper also " Admixtures have the potential to significantly increase concrete’s resistance to frost, which is essential for maintaining the long-term stability and security of concrete structures in cold climates" sounds strange. It is partly the general reference of "Admixtures" but also common knowledge tells us that "admixtures have the potential". The conclusions should do more to highlight the authors' accomplishments relative to what is presently known.

Comments on the Quality of English Language

The attached comments give advices.

Reviewer 2 Report

Comments and Suggestions for Authors

The authors undertook appealing and relevant research, with the goal of expanding the use of C&D waste in modern building materials. The study is well-documented, and the findings are clearly described. 

In this work, the authors proved the effects of several chemical admixtures on the frost resistance of recycled aggregate concrete, which is common knowledge in the field of concrete technology. However, more environmentally friendly ways include employing waste materials to enhance the shortcomings of RAC characteristics rather than costly and commercial chemical admixtures. In this regard, the Introduction and the Conclusion section, at the very least, should be enhanced and supplemented with the state of the art in the field of utilizing recycled and waste materials in the production of low-temperature-resistant concrete. For example, the authors proposed rubber particles (obtained from waste recycling) as one feasible method for improving the frost resistance of RAC. Please outline the advantages and disadvantages of using such materials, as well as their effects on concrete frost resistance, and highlight areas where the assessed chemical admixtures are a superior answer, despite the principles of sustainable development in contemporary civil engineering practice. 

In addition, the following shortcomings should be addressed as well: 

- The use of RCA in composites has been studied for decades, and some basic understanding has been collected. The authors should elaborate on these findings in the Introduction section (cite some articles). This would assist indicate the study's novelty. 

- Some photographs from the conducted experiment would help to visualize the article. The graphical abstract is also a possible choice. 

- Methods related to the characterization of materials should be mentioned and described.

- The impact of air-entraining admixture on RAC frost resistance should be discussed further in the Conclusion section. 

Comments on the Quality of English Language

The English language is quite decent. Minor editing of English language is required.

Reviewer 3 Report

Comments and Suggestions for Authors

Main impression:

The effect of the incorporation of chemical admixtures on frost resistance and microscopic properties of recycled coarse aggregate concrete was evaluated. Coarse recycled concrete aggregates were used to replace natural coarse aggregates on ordinary concrete. Chemical admixtures included antifreeze-type water-reducing admixture, air entraining admixture and antifreeze admixture. The obtained results are interesting and worthy of publication. In general, the abstract clearly outline the study carried out and the main conclusion, although a minor revision is necessary on terminology used. Keywords needs to be reformulated. The introduction section is reasonably presented and can be considered consistent. The materials and experimental methods section need a major revision to provide additional clarifications and information to fully understand the methodology and their impact on results. The results section is reasonably presented, but basically limits itself to presenting the obtained results, with a clear lack of discussion of these results in comparison with those described in the literature. It is strange that the articles presented in the literature review in the introduction section were not used for discussion. Conclusions are consistent and supported by the obtained results, although a minor revision is needed. Finally, the references are correctly presented and reasonably cover the scope of the research done. A list of comments/suggestions to allow the authors to improve the quality of the manuscript is presented below.

Concerning the “Title”:

(1) Page 1, line 4: The reference to “External Additives” is very unusual. It should be “Chemical Admixtures”.

Concerning the “Abstract”:

(2) Page 1, line 10: What type of recycled coarse aggregates were used? It should be “… coarse recycled concrete aggregates ….”, instead of “… recycled coarse aggregate …”.

(3) Page 1, line 11 to 12: It should be “… investigate the effects of antifreeze-type water-reducing admixture, air entraining admixture and antifreeze admixture on the frost resistance ….”, instead of “… investigate the effects of antifreeze water reducers, air entraining agents and antifreeze agents on the frost resistance …”.

(4) Throughout the article, use the word "admixture" instead of "additive" or "agent".

(5) Throughout the article, do not refer to “external additives”. It should be “chemical admixtures” or simply “admixtures”.

Concerning the “Keywords”:

(6) Page 1, line 25: Reconsider the following keywords as a suggestion (max. 10): recycled coarse aggregate concrete; recycled aggregates; chemical admixtures; durability; frost resistance; mechanical properties; mass loss rate; dynamic modulus of elasticity; porosity; microscopy.

Concerning the “1. Introduction”:

(7) Page 1, line 36: It should be “… coarse recycled concrete aggregates ….”, instead of “… recycled coarse aggregates …”

(8) Page 2, line 63 to 65: It is mentioned: “… in contrast to regular aggregates, recycled coarse aggregates inject three distinct types of interfacial transition zones into the freshly created concrete.”. What do you mean by “inject”? Do these interfacial transition zones occur in freshly created concrete or in hardened concrete? Review the sentence and add a brief explanation concerning the “three different types of interfacial transition zones” referred to in the text.

(9) Page 2, line 82: It should be “Çullu and Arslan [30]”, instead of “Arslan et al. [30]”.

Concerning the “2. Materials and Experimental Methods”:

(10) Page 3, line 104: It should be “… a density of 2591 …”, instead of “… an apparent density of 2591 …”.

(11) Page 3, line 112: It should be “… its density is 2618 …”, instead of “… its apparent density is 2618 …”.

(12) Page 3, line 120 to 122: Something is missed in this paragraph!

(13) Page 4, line 124, Table 2: What is FA? Fly Ash was never mentioned previously! It should be included in materials section.

(14) Page 4, Section “2.1. Materials”: What about air entraining admixture? It should be included in materials section.

(15) Page 5, Table 5, Title: Title does not match with values presented. Where is the antifreeze dosage?

(16) Page 5, Table 5: What means sand rate? It is the first time it is mentioned in this paper!

(17) Page 5, Table 5: Show the proportions of all the mixtures produced (including admixtures) associated with the respective abbreviation! This data is necessary to fully understand the experimental programme.

(18) Page 5, Table 5: For the reference mixtures shown on Table 5 the water content is very high. You have mentioned that slump was kept constant within a 220–240 mm range. What is the impact on water dosage on the other produced mixtures when using water reducer admixture and how it affects water to binder ratio? Did this affect frost resistance evaluation?

(19) Page 5, line 150: It should be “… consistency …”, instead of “… flowability …”.

(20) Page 5, line 169: How relative dynamic modulus of elasticity was measured? What equipment was used?

Concerning the “3. Tests Results”

(21) Page 6, line 197, Section title: It should be “3. Results and Discussion”, instead of “3. Tests Results”. This section should not limit to present results and should include also the discussion of such results, by comparison with those described in the literature on similar research.

(22) Page 7, line 227: It is mentioned “… an anti-freeze water-reduction ingredient.”. Avoid the word “ingredient”. Replace it by “admixture”. Proceed accordingly also in line 248.

(23) Page 7, Last two paragraphs: There are many expressions meaning the same! Please uniformize, because it can confuse the reader. For example: “antifreeze-type water-reducing additives”; “anti-freeze–thaw water-lowering additives”; “a water-reducing chemical of the antifreeze type”.

(24) Page 7, line 249: Rewrite the sentence “The water-reducing agent’s water-reducing component …”.

(25) Page 8, line 274: It is mentioned “…the relative kinetic elastic modulus …”. Is it “kinetic” or “dynamic”? Review other similar situations in this article accordingly.

(26) Page 10, line 324: It should be “… the benefits of adding antifreeze chemicals to recycled coarse aggregate concrete …”, instead of “… the benefits of adding antifreeze chemicals to recycled fine aggregate concrete …”

(27) Page 15, line 412: It should be “… Jin et al. [46] …”, instead of “… Weizhun Jin’s [46] …”.

Concerning the “4. Conclusions”

(28) Page 17, line 460 to 464: The description of the scope of the research could be improved, particularly with regard to the tests carried out.

(29) Page 18, line 475: It should be “… relative dynamic …”, instead of “… relative kinetic …”.

(30) Page 18, line 481 to 484: Revise the sentence, as the last two lines are unintelligible.

Concerning the “References”:

Nothing to report.

Comments on the Quality of English Language

Moderate editing of English language required.

Round 2

Reviewer 1 Report

Comments and Suggestions for Authors

The authors have addressed the review comments. The paper can be recommended for publication.

Comments on the Quality of English Language

Fine

Author Response

Comments 1:The authors have addressed the review comments. The paper can be recommended for publication.

Response 1:Thank you very much for taking the time to review this manuscript. And your very professional and valuable comments on this article, which has been very much enhanced by your guidance. It has been a very rewarding experience for the writer to have received your comments, and I thank you once again for your kind guidance.I have upgraded Quality of English Language

Reviewer 2 Report

Comments and Suggestions for Authors

No additional issues were detected.

Author Response

Comments 1:No additional issues were detected. 

Response 1:Thank you very much for taking the time to review this manuscript. And your very professional and valuable comments on this article, which has been very much enhanced by your guidance. It has been a very rewarding experience for the writer to have received your comments, and I thank you once again for your kind guidance. 

Reviewer 3 Report

Comments and Suggestions for Authors

The authors have performed a revision of the manuscript following the reviewers' instructions, which has provided an improvement in the overall quality of the article. The new title presented by the authors can be considered more appropriate to the research carried out. However, not all the requested revision was entirely presented, which makes it difficult to effectively evaluate the research programme and how it affects the obtained results. In addition, some small gaps were detected in the new text included in the revision process, which can easily be overcome by the authors. In the reviewer's opinion, the article can only be considered suitable to the journal’s required standard and accepted for publication if a minor revision of the detected gaps is provided, as listed below.

(1) Page 1, Line 18: It should be “… the admixtures have …”, instead of “… the Admixture have …”.

(2) Page 1, Line 24: It should be “… antifreeze admixture had …”, instead of “… antifreeze Admixture had …”.

(3) Page 2, Line 48: It should be “… Elansary et al. [18]. The results …”, instead of “… Elansary AA et al. [18] The results …”.

(4) Page 2, Lines 50 to 51: It should be “… than when using normal coarse aggregate (NCA), while the use of 100% CARR resulted in worse mechanical properties than with NCA.”, instead of “… than NCA, while the use of 100% CARR resulted in worse mechanical properties than NCA.”.

(5) Page 2, Line 74: “… is more difficult, because there are three distinct …”, instead of “… is more difficult because, There are three distinct …”.

(6) Page 2, Lines 94 to 95: It should be “… Çullu and Arslan [30] …”, instead of “… Çullu and Arslan. [30] …”.

(7) Page 3, Line 105: It should be “… and reduce labor intensity.”, instead of “… and Reduce labor intensity.”.

(8) Page 4, Table 1: It should be “Amount of Residue [g]”, instead of “Amount of Residue /g”.

(9) Page 4, Lines 142 to 144: It should be “Three different admixtures were also used: the antifreeze-type water-reducing admixture, which is a translucent liquid containing 30% solids, an air-entraining agent based on alkylbenzene sulphonate, which is a translucent liquid, and an antifreeze agent belongs to …”. instead of “For admixtures, antifreeze-type water-reducing admixture is a translucent liquid containing 30% solids. Air-entraining agent is alkyl benzene sulfonate, translucent liquid. Antifreeze agent belongs to …”.

(10) Page 5, Table 5 title: As stated from line 152 to 153, “The amounts of various materials per cubic meter for the planned concrete with water–cement ratios of 0.56 and 0.41 are displayed in Table 5”. Therefore, as previously requested, the title of Table 5 should be revised. It should be “Mix proportions per cubic metre of the reference concretes.”, instead of “Addition of antifreeze admixture for recycled concrete mix design.”.

(11) Page 5, Table 5, Footnote: is it “sand ratio” or “sand rate”? Was this ‘sand rate’ used in analysing the results? If it hasn't been used, what is the need to include this parameter in this table?

(12) Page 5, Mix proportions: As previously request, please add the proportions of all the mixtures produced (including admixtures dosages) associated with the respective abbreviation! This data is necessary to fully understand the experimental programme.

(13) Page 5, Lines 173 to 175, Configuration Process and Maintenance Conditions: It is mentioned that “… the recycled coarse aggregate was mixed directly with water without pre-saturated surface dry treatment…”. How was the water absorption of the recycled aggregates controlled? An additional portion of water was added to compensate for recycled aggregates absorption? If the recycled aggregates absorb part of the mixing water, this means that the effective water dosage decreases and this affects the water/binder ratio and consistency. Was the loss of consistency compensated by the water reducer admixture? How can you guarantee that the effective water/binder ratio is constant? The procedures are not clear for the reader. Please include additional explanation.

(14) Page 18, Conclusion no 3: Revise all the paragraph of conclusion no 3, as it is still very confusing for the reader...

Comments on the Quality of English Language

Minor editing of English language required.
